# Comparative Analysis of Bacterial Community Structures in Earthworm Skin, Gut, and Habitat Soil across Typical Temperate Forests

**DOI:** 10.3390/microorganisms12081673

**Published:** 2024-08-14

**Authors:** Kang Wang, Ning Yuan, Jia Zhou, Hongwei Ni

**Affiliations:** 1School of Geographical Sciences, Harbin Normal University, Harbin 150025, China; kangwang2021@outlook.com (K.W.); yuanning@iga.ac.cn (N.Y.); 2Heilongjiang Academy of Forestry, Harbin 150081, China

**Keywords:** earthworm, bacteria, diversity analysis, mantel test, environmental factors

## Abstract

Earthworms are essential components in temperate forest ecosystems, yet the patterns of change in earthworm-associated microbial communities across different temperate forests remain unclear. This study employed high-throughput sequencing technology to compare bacterial community composition and structure in three earthworm-associated microhabitats (skin, gut, and habitat soil) across three typical temperate forests in China, and investigated the influence of environmental factors on these differential patterns. The results indicate that: (1) From warm temperate forests to cold temperate forests, the soil pH of the habitat decreased significantly. In contrast, the physicochemical properties of earthworm skin mucus exhibited different trends compared to those of the habitat soil. (2) Alpha diversity analysis revealed a declining trend in Shannon indices across all three microhabitats. (3) Beta diversity analysis revealed that the transition from warm temperate deciduous broad-leaved forest to cold temperate coniferous forest exerted the most significant impact on the gut bacterial communities of earthworms, while its influence on the skin bacterial communities was comparatively less pronounced. (4) *Actinobacteria* and *Proteobacteria* were the predominant phyla in earthworm skin, gut, and habitat soil, but the trends in bacterial community composition differed among the three microhabitats. (5) Mantel tests revealed significant correlations between bacterial community structures and climatic factors, physicochemical properties of earthworm habitat soil, and physicochemical properties of earthworm skin mucus. The findings of this study offer novel perspectives on the interplay between earthworms, microorganisms, and the environment within forest ecosystems.

## 1. Introduction

Earthworms, functioning as key ecosystem engineers in soil ecosystems, play a crucial role in maintaining soil health and promoting ecosystem functions [1]. These invertebrates exhibit complex interdependencies with soil and environmental factors, which are essential for understanding and managing forest ecosystems [2].

The impact of earthworms on soil is multifaceted and can be categorized into direct and indirect effects. Direct influences are primarily achieved through their feeding and excretion activities. Earthworms ingest soil and organic matter, mixing and enriching nutrients during digestion, and subsequently excreting them in the form of earthworm casts [3]. This process not only accelerates the decomposition of organic matter but also enhances the bioavailability of soil nutrients [4]. For instance, earthworm activity can significantly increase the content of key elements such as nitrogen, phosphorus, and potassium in the soil [5]. Furthermore, the burrowing and tunneling behaviors of earthworms improve the physical structure of the soil [6,7]. The network of channels they create in the soil increases soil porosity, facilitating the circulation of water and air, thereby enhancing soil aeration and permeability [8]. These modifications to soil structure are conducive to plant root growth [9]. The indirect effects of earthworms are primarily manifested through their complex interactions with microbial communities. The diverse and abundant microbial populations present in earthworm skin and gut play crucial roles in processes such as organic matter decomposition, nutrient transformation, and pathogen suppression [10,11,12,13]. For instance, microorganisms in the earthworm gut secrete various enzymes that accelerate the degradation of recalcitrant organic compounds such as cellulose and hemicellulose [14]. Concurrently, earthworm activity can alter the composition and function of soil microbial communities, thereby influencing the biogeochemical cycles of the entire soil ecosystem [15,16]. However, the activity and distribution of earthworms are also strongly influenced by environmental factors. Climatic conditions (such as temperature and precipitation), soil physicochemical properties (including pH and organic matter content), and vegetation types directly or indirectly affect earthworm survival, reproduction, and activity [17,18,19,20,21]. This interplay creates a complex feedback loop, rendering the earthworm–soil–environment system an intricately interconnected entity.

Temperate forests, as one of the most widely distributed terrestrial ecosystems on Earth, play an irreplaceable role in global carbon cycling, biodiversity conservation, and climate regulation [22,23]. However, different types of temperate forests exhibit significant variations in climatic conditions, vegetation composition, and soil characteristics, which may profoundly influence earthworms and their associated microbial communities [24]. Concurrently, the quality and quantity of litter in various forest types affect soil organic matter content and quality [25,26], subsequently impacting earthworm food sources and microbial community composition. Although previous studies have explored the distribution patterns and ecological functions of earthworms in temperate forests [27,28,29], systematic comparative research on soil, skin, and gut microbial communities associated with earthworms across different temperate forest types remains limited. In particular, our understanding of how the transition from warm temperate deciduous broadleaf forests to cold temperate coniferous forests influences earthworm-associated microbial communities is still constrained.

This study aims to compare the composition and structure of bacterial communities in the skin, gut, and habitat soil of *Eisenia nordenskioeldi* across three typical temperate forests in China: warm temperate deciduous broad-leaved forest, temperate broad-leaved red pine forest, and cold temperate coniferous forest. Additionally, we investigate the influence of climatic factors, soil physicochemical properties, and earthworm skin mucus characteristics on the observed differentiation patterns. We hypothesize that: (1) Bacterial communities in earthworm skin, gut, and habitat soil will exhibit distinct response patterns from warm temperate deciduous broad-leaved forest to cold temperate coniferous forest; (2) These differences are primarily influenced by climatic factors, soil physicochemical properties, and earthworm skin mucus characteristics.

## 2. Materials and Methods

### 2.1. Ethics Statement

All animal experiments were carried out under the guidelines approved by the Institutional Animal Care and Use Committee (IACUC). The proposed study protocols were approved by the Laboratory Animal Care Committee of Harbin Normal University (No. HNUARIA2024015).

### 2.2. Sample Collection

#### 2.2.1. Study Site

This study selected three representative temperate forest ecosystems in northern China as research areas: the Dongling Mountains warm temperate deciduous broad-leaved forest (DL) (39°58′ N, 115°26′ E), the Changbai Mountains temperate broad-leaved red pine forest (CB) (42°38′ N, 128°08′ E), and the Greater Hinggan Mountains cold temperate coniferous forest (GH) (51°49′ N, 122°59′ E). These three research areas represent warm temperate, mid-temperate, and cold temperate forest ecosystems, respectively, distributed along a latitudinal gradient from south to north (Figure 1). The study plots in these three areas cover 20 hectares, 25 hectares, and 25 hectares, respectively. All studies and determinations pertaining to earthworm skin, gut, and habitat soil were conducted across the three distinct forest ecosystems.

#### 2.2.2. Experiment Design

In three representative temperate forest ecosystems, ten 20 m × 20 m sampling plots were randomly established, with a minimum distance of 50 m between adjacent plots. Within each sampling plot, nine 1 m × 1 m quadrats were selected using an S-shaped sampling method for earthworm collection. Mature specimens of *Eisenia nordenskioeldi* of uniform size were collected. The collection process involved manual sieving of the litter layer and organic layer, as well as excavation and examination of the top 0–10 cm of mineral soil. Identical collection methods and intensity were applied to all quadrats.

Soil samples were collected using an 8 cm diameter soil auger from the vicinity of each earthworm sampling point. The samples were transferred to sterile, sealed bags and transported in a portable refrigerator maintained at 4 °C. Upon arrival at the laboratory, samples were immediately sieved through a 2 mm mesh to remove stones and plant debris. The sieved soil samples were then divided into two portions: one portion was air-dried at room temperature for subsequent physical and chemical analyses, while the other portion was stored at −80 °C for future microbial analysis.

#### 2.2.3. Bacterial Collection from Earthworm Skin

This study employed a sampling methodology adapted from [30,31,32]. Throughout the sampling process, strict aseptic procedures were adhered to, including changing sterile gloves for each earthworm skin sample collection and pre-sterilizing all instruments and materials. Earthworm skin samples were collected using sterile flocked swabs (Kunronda). The swab was rolled from one end of the earthworm to the other with moderate pressure, repeating the process 10 times. Each swab was used to sample 5 earthworms, with a total of 5 swab samples collected per site. Post-sampling, swabs were immediately placed in sterile centrifuge tubes and stored at −80 °C for subsequent DNA extraction. Sampled earthworms were preserved in anhydrous ethanol and frozen at −80 °C for future gut sample extraction [33]. A total of 25 earthworms per site were collected for both skin and gut sampling. The remaining collected earthworms were placed in perforated plastic boxes along with their native soil and transported to the laboratory for subsequent skin mucus collection experiments.

#### 2.2.4. Earthworm Skin Mucus Collection

This study employed a methodology adapted from [34,35]. Mature earthworms of uniform size were selected and washed five times with deionized water, followed by surface drying with filter paper. The processed earthworms were then placed in Petri dishes lined with moistened filter paper, with two individuals per dish. The dishes were transferred to a constant temperature incubator set at 22 °C and maintained in darkness for 48 h to ensure complete evacuation of gut contents. To prevent coprophagy, the filter paper was replaced twice every 24 h. Following the evacuation period, the earthworms were again washed with deionized water and surface-dried. Subsequently, 50 processed earthworms were transferred to a 500 mL beaker and evenly covered with 200 g of quartz sand (GR, Karan, Xi’an, China) with a particle size of 40–70 mesh. The beaker was then incubated at 22 °C in darkness for an additional 24 h. Post-incubation, the beaker, earthworms, and quartz sand were washed five times with 300 mL of deionized water. The wash solution was collected, filtered, and the filtrate was stored at −20 °C for subsequent analysis.

#### 2.2.5. Earthworm Gut Collection

Earthworm specimens were removed from anhydrous ethanol and rinsed three times with sterile water. Aseptic dissection was performed on an ultra-clean bench to extract the earthworm gut contents. Employing the same pooling strategy as used for the earthworm skin samples, gut contents from five corresponding earthworms were combined to form a single sample. A total of five earthworm gut samples were collected and immediately stored at −80 °C for subsequent DNA extraction [33].

### 2.3. Physicochemical Properties of Soil and Skin Mucus

The pH of soil and skin mucus was determined in a 1:5 (*w*:*v*) suspension using a pH meter. Total carbon (TC) and total nitrogen (TN) contents of soil and skin mucus were measured using an elemental analyzer (Vario max CN, Elementar, Hanau, Germany) via dry combustion method. The carbon-to-nitrogen ratio (C/N) was calculated by dividing TC by TN. Total phosphorus (TP) in soil and skin mucus was extracted using H_2_SO_4_-HClO_4_ digestion, followed by determination using the molybdenum blue colorimetric method. Total potassium (TK) in soil and skin mucus was determined using flame photometry. Nitrate nitrogen (NO_3_^−^-N) and ammonium nitrogen (NH_4_^+^-N) contents in soil and skin mucus were measured using a continuous flow analyzer (SAN Plus, Skalar, Erkelenz, Germany). Soil temperature (ST) at each sampling point was measured using a right-angle soil thermometer. Mean annual temperature (MAT) and mean annual precipitation (MAP) data were obtained from the National Science and Technology Infrastructure platform.

### 2.4. DNA Extraction of Bacterial Communities

Microbial DNA was extracted from litter samples using the E.Z.N.A. Soil DNA Kit (Omega Bio-Tek, Norcross, GA, USA). DNA concentration and quality were assessed by 1% agarose gel electrophoresis (1× TAE buffer, 100 V, 20 min). PCR amplification of the V3-V4 hypervariable regions of the bacterial 16S rRNA gene was performed using primers 338F (5′-ACTCCTACGGGAGGCAGCAG-3′) and 806R (5′-GGACTACHVGGGTWTCTAAT-3′) on a GeneAmp 9700 thermal cycler (ABI, Foster City, CA, USA). The amplification protocol consisted of initial denaturation at 94 °C for 4 min, followed by 25 cycles of 94 °C for 30 s, 55 °C for 30 s, and 72 °C for 1 min, with a final extension at 72 °C for 10 min. Amplification products were purified after 2% agarose gel electrophoresis. Barcodes (8 base pairs) were used to distinguish samples and were removed during subsequent sequence processing. Amplicon libraries were constructed and sequenced on the Illumina Novaseq 6000 platform by Majorbio Bio-Pharm Technology Co., Ltd. (Shanghai, China). Sequence data were processed using the DADA2 pipeline in QIIME2 (2023.2) for quality control, denoising, and clustering into Amplicon Sequence Variants (ASVs). Taxonomic annotation of 16S sequences was performed using the Silva138 database. All samples were rarefied to an equal sequencing depth (10,869 sequences) for analysis. During sample processing, one earthworm skin microbial sample failed detection, resulting in four replicates (*n* = 4) for earthworm skin mucus and corresponding microbial samples, while other sample types maintained five biological replicates (*n* = 5). The raw sequence data were uploaded to the SRA database at NCBI with number PRJNA1139974.

### 2.5. Statistical Analyses

Alpha diversity indices were computed utilizing the ‘estimateR()’ and ‘diversity()’ functions from the vegan package to assess bacterial community richness and diversity. The Kruskal–Wallis test was performed using the ‘kruskal.test()’ function to evaluate significant differences in soil physicochemical properties, earthworm skin mucus physicochemical properties, α-diversity indices, and bacterial abundance across different temperate forest types. Non-metric multidimensional scaling (NMDS) analysis, based on Bray–Curtis distances, was conducted using the ‘metaMDS()’ function from the vegan package to visualize differences in bacterial community structures among temperate forest types. To further investigate differences in bacterial community structures, distance matrices were computed using the ‘vegdist()’ function from the vegan package. Subsequently, ANOSIM and PERMANOVA analyses were performed using the ‘anosim()’ and ‘adonis()’ functions, respectively, to test for significant differences in bacterial community composition across temperate forest types. Mantel tests were conducted using the ‘mantel()’ function to explore correlations between climatic factors, soil physicochemical properties, earthworm skin mucus physicochemical properties, and bacterial community structures. Bar charts were generated using the ‘barplot()’ function, while box plots and stacked bar charts were created using the ‘geom_bar()’ and ‘geom_boxplot()’ functions from the ggplot2 package. All analyses were performed using R version 4.2.3.

## 3. Results

### 3.1. Physicochemical Properties of Earthworm Habitat Soils and Skin Mucus

Significant differences in soil physicochemical properties of earthworm habitats were observed across the three temperate forest types (Figure 2). Firstly, soil pH exhibited a significant decreasing trend from DL to GH (*p* < 0.001). Secondly, CB consistently demonstrated the highest overall soil nutrient content, with significantly higher levels of TN, TC, TP, NO_3_^−^-N, and NH_4_^+^-N compared to the other two forest types (*p* < 0.001). In contrast, DL exhibited intermediate soil nutrient levels, while GH showed the lowest concentrations of TN, TC, and TP. Notably, the patterns of TK content and C/N differed from other soil physicochemical properties across the three temperate forest types. GH displayed the highest TK content and C/N ratio, which were significantly higher than those in the other two forest types (*p* < 0.001).

The physicochemical properties of earthworm epidermal mucus exhibited trends distinct from those observed in soil characteristics across different forest types (Figure 3). Specifically, several parameters of the earthworm epidermal mucus, including TN, TC, TK, and NH_4_^+^-N, were significantly lower in the GH forest compared to DL and CB forests (*p* < 0.05). Conversely, the nitrate nitrogen NO_3_^−^-N content in the earthworm epidermal mucus was significantly higher in GH than in DL (*p* < 0.05). The TP content in the mucus was significantly higher in CB compared to the other two forest types (*p* < 0.01). Despite these variations in physicochemical properties of earthworm epidermal mucus across the three temperate forest types, no significant differences were observed in pH values or carbon-to-nitrogen ratios (C/N) among the forest types (*p* > 0.05).

Furthermore, we assessed key climatic parameters across the three temperate forest types, including MAT, MAP, and ST in earthworm habitats (Figure 4). A gradual decrease in both MAT and ST was observed along the gradient from DL to GH. The highest MAP was recorded in CB, while GH exhibited the lowest MAP among the three forest types.

### 3.2. Bacterial Community Diversity and Structure

#### 3.2.1. Bacterial Community Alpha Diversity

The bacterial community α-diversity in earthworm skin, gut, and habitat soil exhibited variations across the three temperate forest types (Figure 5). Analysis of the Chao1 index revealed no significant differences in earthworm habitat soil and gut bacterial communities among the three temperate forest types (*p* > 0.05). However, the Chao1 index for earthworm skin bacterial communities was significantly lower in GH compared to CB (*p* < 0.001). Examination of the Shannon index showed that bacterial diversity in earthworm habitat soil was significantly lower in GH than in DL (*p* < 0.05). A similar trend was observed for earthworm skin bacterial communities, with the Shannon index being significantly lower in GH compared to both DL and CB (*p* < 0.001). Furthermore, the Shannon index for earthworm gut bacterial communities was significantly lower in CB than in DL (*p* < 0.05). Overall, a gradual decrease in the Shannon index was observed from DL to GH across earthworm skin, gut, and habitat soil bacterial communities.

#### 3.2.2. Bacterial Community Beta Diversity

Non-metric multidimensional scaling (NMDS) ordination revealed significant differences in bacterial community structures of earthworm skin, gut, and habitat soil across the three temperate forest types (PERMANOVA, *p* < 0.001) (Figure 6). The transition from DL to GH significantly influenced the bacterial community structures in earthworm skin, gut, and habitat soil (ANOSIM and PERMANOVA, *p* < 0.05) (Table 1). PERMANOVA analysis indicated that the transition from DL to GH exerted the most substantial influence on the bacterial community structure of earthworm gut (R^2^ = 0.471–0.596), followed by the earthworm habitat soil (R^2^ = 0.326–0.38), while the impact on the bacterial community structure of earthworm skin was comparatively less pronounced (R^2^ = 0.322–0.329) (Table 1).

#### 3.2.3. Bacterial Community Composition

In the three temperate forests investigated, *Actinobacteria* and *Proteobacteria* were the predominant bacterial phyla in the earthworm skin, gut, and habitat soil. Additionally, *Acidobacteria* and *Chloroflexi* exhibited relatively high abundances in the earthworm habitat soil and skin. However, the bacterial community composition in the earthworm gut displayed significant differences compared to that of the soil and skin, with *Firmicutes* and *Verrucomicrobia* presenting higher relative abundances in the earthworm gut (Figure 7).

In earthworm skin, gut, and habitat soil, both CB and GH demonstrated significantly higher relative abundances of *Verrucomicrobia* compared to DL (*p* < 0.05). Specifically, CB exhibited a statistically significant increase in the relative abundance of *Verrucomicrobia* across all three microhabitats when compared to DL. GH displayed a similar trend relative to DL. However, no statistically significant differences (*p* > 0.05) were observed between CB and GH in terms of *Verrucomicrobia* relative abundance in any of the examined microhabitats (earthworm skin, gut, or habitat soil). Compared to DL and CB, GH exhibited significantly lower relative abundances of *Chloroflexi* in both earthworm skin and gut (*p* < 0.05). However, no statistically significant differences (*p* > 0.05) were observed in the relative abundance of *Chloroflexi* in the earthworm habitat soil among the three temperate forest types (DL, CB, and GH). In earthworm habitat soil and skin, the relative abundance of *Proteobacteria* gradually increased (*p* < 0.05). However, the bacterial community composition in the earthworm gut exhibited a different trend compared to the habitat soil and skin. The relative abundance of *Acidobacteria* showed a significant upward trend (*p* < 0.01), whereas the relative abundance of *Myxococcota* decreased significantly (*p* < 0.05) (Figure 8).

### 3.3. Environmental Factors Influencing Bacterial Communities

Mantel test results revealed complex relationships between environmental factors and bacterial community structures in earthworm skin, gut, and habitat soil (Figure 9). Firstly, climatic factors, including MAT, MAP, and ST of earthworm habitats, exhibited significant correlations with bacterial community structures in earthworm skin, gut, and habitat soil (*p* < 0.01). Secondly, soil physicochemical properties not only significantly correlated with bacterial community structures in earthworm habitat soil (*p* < 0.05) but also demonstrated stronger correlations with bacterial community structures in earthworm gut (*p* < 0.01). Furthermore, upon analyzing the physicochemical characteristics of earthworm skin mucus, we found that four key indicators—TN, TC, TP, and TK—also showed significant correlations with bacterial community structures in earthworm skin (*p* < 0.05). Collectively, these results suggest that climatic factors, soil properties, and the physiological characteristics of earthworms themselves may exert substantial influences on their associated microbial community structures.

## 4. Discussion

### 4.1. Effects of Temperate Forest Type Transition on Bacterial Community Diversity in Earthworm-Associated Microhabitats

We observed significant differences in the physicochemical properties of earthworm habitat soil and skin mucus across the three temperate forest types. These variations are likely driven by changes in climatic factors and vegetation types [36]. The physicochemical properties of earthworm skin mucus exhibited different trends compared to those of habitat soil, suggesting that earthworms may adapt to diverse environmental conditions by regulating the secretion of skin mucus [37,38]. This discovery provides novel insights into the adaptive mechanisms of earthworms in diverse temperate forest ecosystems.

We observed significant alterations in the diversity and structure of bacterial communities associated with earthworm skin, gut, and habitat soil along the transition from DL to GH. Alpha diversity analysis revealed a gradual decrease in bacterial community diversity from DL to GH, potentially due to reduced temperature and nutrient availability limiting microbial growth and metabolism [39].

Beta diversity analysis revealed disparities in bacterial community structures across different temperate forest ecosystems. Notably, the transition from DL to GH exhibited the most pronounced impact on the bacterial community structure within the earthworm gut. This indicates that earthworm gut bacterial communities are more sensitive to environmental changes compared to those in habitat soil and skin. This differentiated response pattern represents a novel finding. This heightened sensitivity may be attributed to the close association between gut bacterial communities and the host’s digestive and nutrient absorption processes [40], their higher metabolic capacity [12,41], and potential influences from changes in food sources [42]. This discovery expands our understanding of earthworm–microbe–environment interactions, underscoring the potential value of earthworm gut microbiomes as indicators of environmental change.

### 4.2. Impact of Temperate Forest Type Transition on Bacterial Community Composition in Earthworm-Associated Microhabitats

Regarding bacterial community composition, we found that *Actinobacteria* and *Proteobacteria* were the predominant phyla in earthworm skin, gut, and habitat soil [43,44]. The bacterial community composition in earthworm gut microbiomes exhibited significant differences compared to those in earthworm skin and habitat soil. *Acidobacteria* and *Chloroflexi* demonstrated higher relative abundances in earthworm skin and habitat soil compared to the earthworm gut [45]. *Acidobacteria*, *Actinobacteria*, and *Proteobacteria* were found to be predominant in forest soils [46,47,48]. The higher carbon-to-nitrogen ratio in earthworm guts resulted in substantially lower abundances of *Acidobacteria* compared to habitat soil [49]. *Firmicutes* and *Verrucomicrobia* displayed higher relative abundances in earthworm guts compared to earthworm skin and habitat soil. These phyla are recognized as dominant taxa in earthworm gut microbiomes [12,41,49,50,51,52,53]. The earthworm gut provides a near-anaerobic microenvironment [33,54], which favors the growth of anaerobic or facultative anaerobic bacteria [55]. Certain bacteria within the *Firmicutes* are well-adapted to thrive in such conditions [56].

Significant changes in the relative abundances of certain bacterial phyla were observed across the gradient from DL to GH. For instance, compared to DL, both CB and GH exhibited significantly higher relative abundances of *Verrucomicrobia* in earthworm skin, gut, and habitat soil (*p* < 0.05). This trend may be attributed to the preference of *Verrucomicrobia* for oligotrophic environments [57]. Previous studies have shown that the abundance of *Actinobacteria* in acidic soils positively correlates with pH [58,59]. The gradual decrease in pH of earthworm habitat soil from DL to GH may explain the reduction in the relative abundance of *Actinobacteria* in this microhabitat. From DL to GH, the relative abundance of *Acidobacteria* in earthworm gut showed a gradually increasing trend, while there was no significant difference in its relative abundance in earthworm skin and habitat soil. *Acidobacteria* are considered slow-growing oligotrophs [60] capable of utilizing complex carbon substrates [61]. The increase in the relative abundance of *Acidobacteria* in the earthworm gut may reflect changes in the host’s digestive strategy, allowing *Acidobacteria* to gain an advantage in the gut of GH earthworms and thus play a more important role in the host’s digestive process.

The Mantel test results further corroborated significant correlations between bacterial community structures and climatic factors, physicochemical properties of earthworm habitat soil, and skin mucus. This aligns with existing research indicating that changes in environmental factors significantly influence the composition and structure of soil and host-associated microbial communities [55]. This finding is also consistent with Szoboszlay et al.’s study, which demonstrated the sensitivity of soil microbial community structures to environmental changes [62]. Concurrently, we cannot overlook the role of earthworms themselves in regulating the physicochemical properties of habitat soil and shaping soil bacterial community structures [7,28,63,64,65,66]. The physicochemical properties of earthworm mucus also correlated with skin bacterial community structures, suggesting that earthworm mucus may provide a unique microenvironment and nutrient source for skin microbiota. Previous studies have shown that earthworm mucus contains carbohydrates, proteins, and amino acids [37,67,68], providing favorable conditions for microbial growth [34,69,70,71]. Furthermore, the strong correlation between earthworm gut bacterial community structures and the physicochemical properties of habitat soil reflects that some gut bacteria may originate from ingested and digested soil [41,72,73,74,75]. Bacteria in earthworm habitat soil and gut exert mutual influences, maintaining a dynamic equilibrium between the two systems [76]. Furthermore, studies have demonstrated that the denitrifying bacteria present in earthworm gut microbiomes originate from ingested soil microorganisms. The unique microenvironment of the earthworm gut facilitates the growth and metabolic activities of these bacteria [77,78,79].

## 5. Conclusions

This study compared the earthworm-associated bacterial communities in three typical temperate forests in China and validated our two main hypotheses: (1) The transition from warm temperate deciduous broad-leaved forest to cold temperate coniferous forest significantly influenced the bacterial community structure in earthworm skin, gut, and habitat soil, but the extent of the impact varied among microhabitats. The earthworm gut bacterial community was the most affected, while the skin bacterial community was relatively less affected. (2) These differences were primarily influenced by climatic factors, soil physicochemical properties, and the properties of earthworm skin mucus. Mantel test results indicated that these three types of factors were significantly correlated with the bacterial community structure in different microhabitats. This study deepens our understanding of earthworm-associated bacterial communities in temperate forest ecosystems.

## Figures and Tables

**Figure 1 microorganisms-12-01673-f001:**
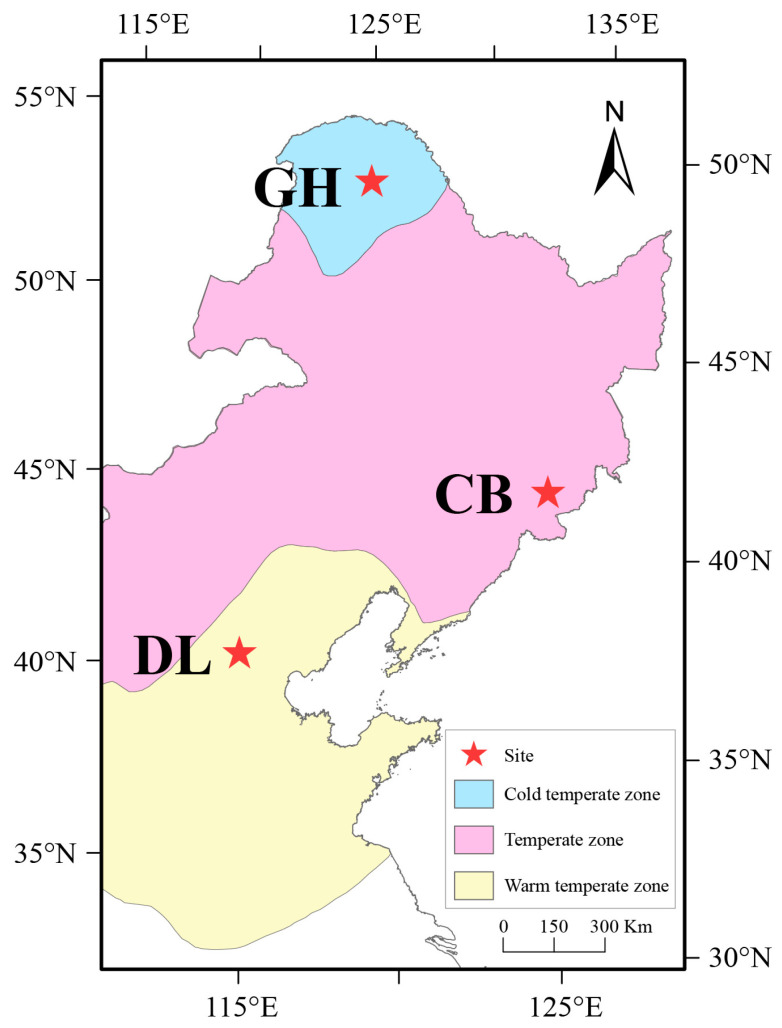
Distribution map of the study plot. DL, Dongling Mountains Warm temperate deciduous broad-leaved forest; CB, Changbai Mountains Temperate broad-leaved red pine forest; GH, Greater Hinggan Mountains Cold temperate coniferous forest.

**Figure 2 microorganisms-12-01673-f002:**
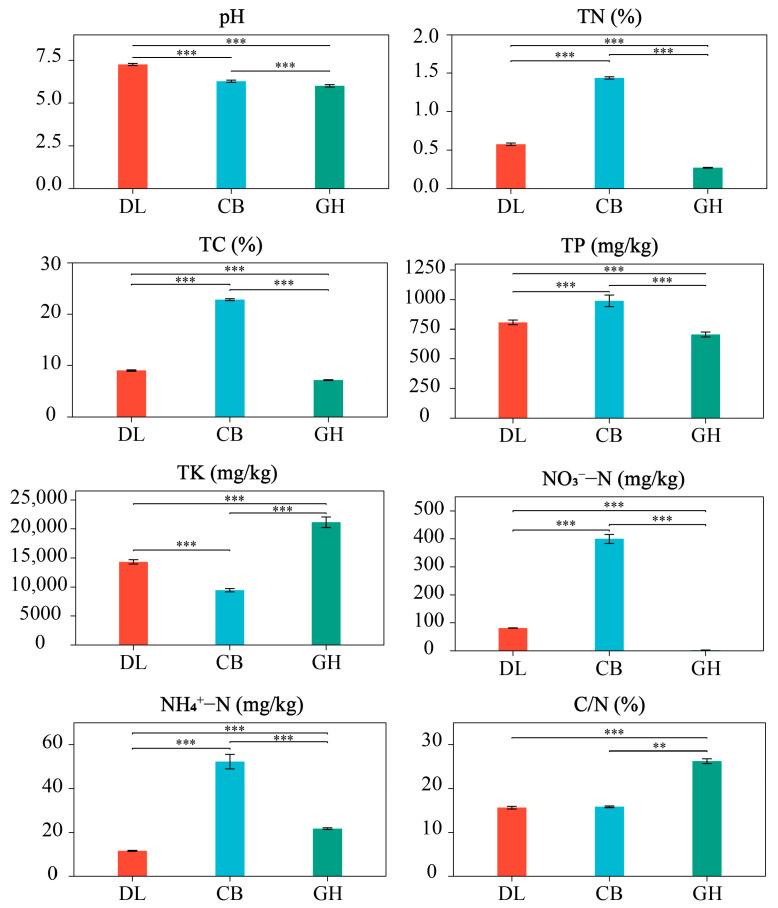
Soil physicochemical properties in three temperate forest types. DL, Dongling Mountains Warm temperate deciduous broad-leaved forest; CB, Changbai Mountains Temperate broad-leaved red pine forest; GH, Greater Hinggan Mountains Cold temperate coniferous forest; TN, total nitrogen; TC, total carbon; TP, total phosphorus; TK, total potassium; NO_3_^−^-N, nitrate nitrogen; NH_4_^+^-N, ammonium nitrogen; C/N, carbon–nitrogen ratio. Significant differences are indicated by asterisks based on the nonparametric Kruskal–Wallis test: ** *p* < 0.01 and *** *p* < 0.001.

**Figure 3 microorganisms-12-01673-f003:**
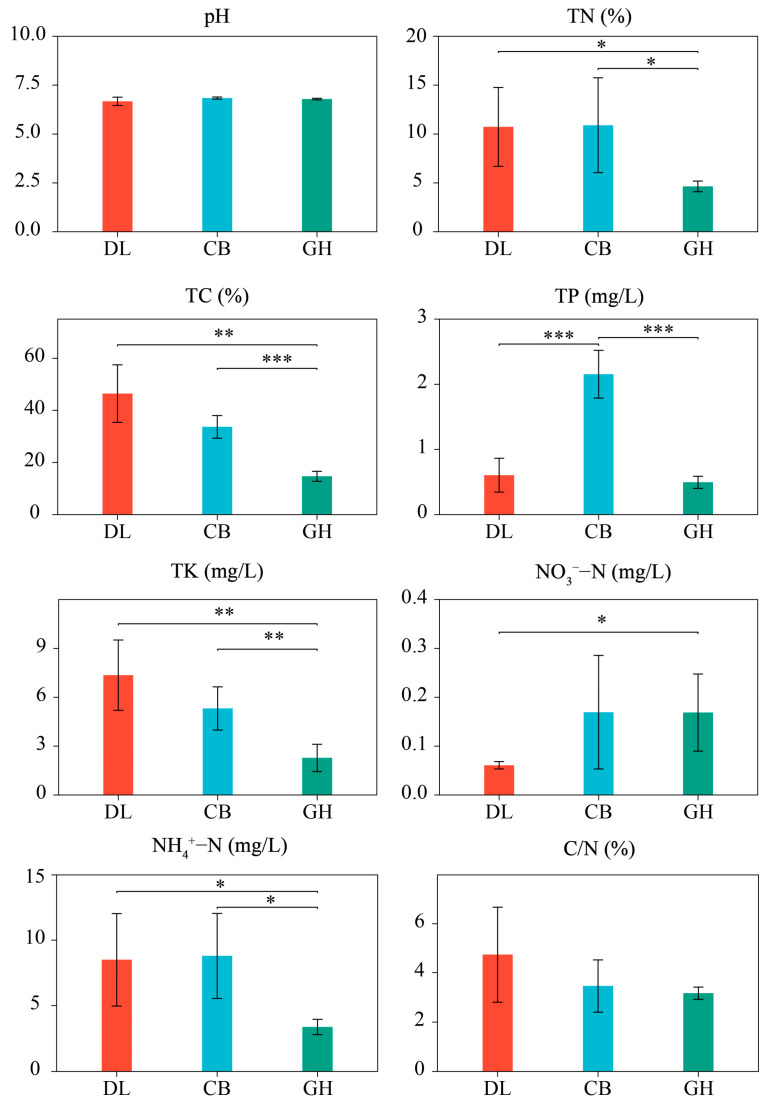
Physicochemical properties of earthworm epidermal mucus in three temperate forest types. DL, Dongling Mountains Warm temperate deciduous broad-leaved forest; CB, Changbai Mountains Temperate broad-leaved red pine forest; GH, Greater Hinggan Mountains Cold temperate coniferous forest; TN, total nitrogen; TC, total carbon; TP, total phosphorus; TK, total potassium; NO_3_^−^-N, nitrate nitrogen; NH_4_^+^-N, ammonium nitrogen; C/N, carbon–nitrogen ratio. Significant differences are indicated by asterisks based on the nonparametric Kruskal–Wallis test: * *p* < 0.05, ** *p* < 0.01, and *** *p* < 0.001.

**Figure 4 microorganisms-12-01673-f004:**
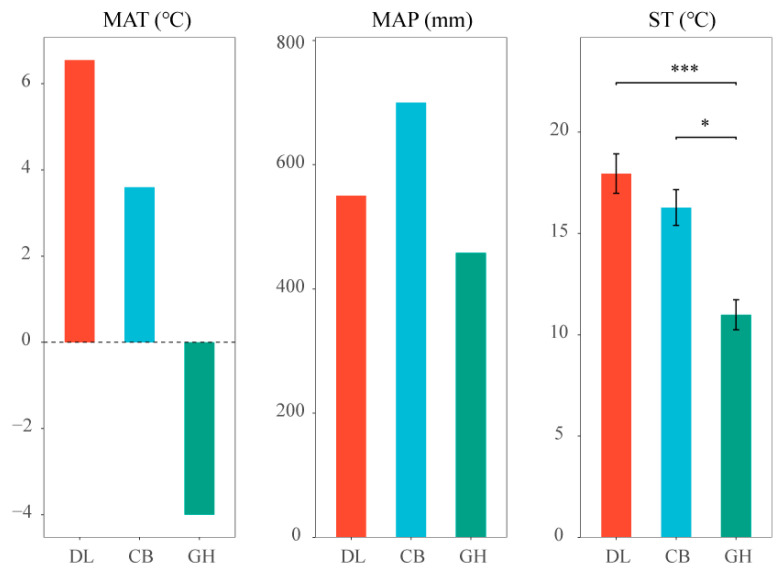
MAT, MAP, and ST in three temperate forest types. DL, Dongling Mountains Warm temperate deciduous broad-leaved forest; CB, Changbai Mountains Temperate broad-leaved red pine forest; GH, Greater Hinggan Mountains Cold temperate coniferous forest; MAT, mean annual temperature; MAP, mean annual precipitation; ST, soil temperature. Significant differences are indicated by asterisks based on the nonparametric Kruskal–Wallis test: * *p* < 0.05 and *** *p* < 0.001.

**Figure 5 microorganisms-12-01673-f005:**
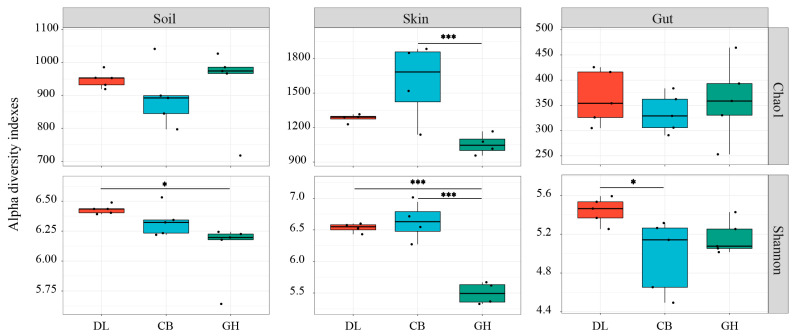
Bacterial community α-diversity in earthworm habitat soil, skin, and gut across three temperate forest types. DL, Dongling Mountains Warm temperate deciduous broad-leaved forest; CB, Changbai Mountains Temperate broad-leaved red pine forest; GH, Greater Hinggan Mountains Cold temperate coniferous forest. Significant differences are indicated by asterisks based on the nonparametric Kruskal–Wallis test: * *p* < 0.05 and *** *p* < 0.001.

**Figure 6 microorganisms-12-01673-f006:**
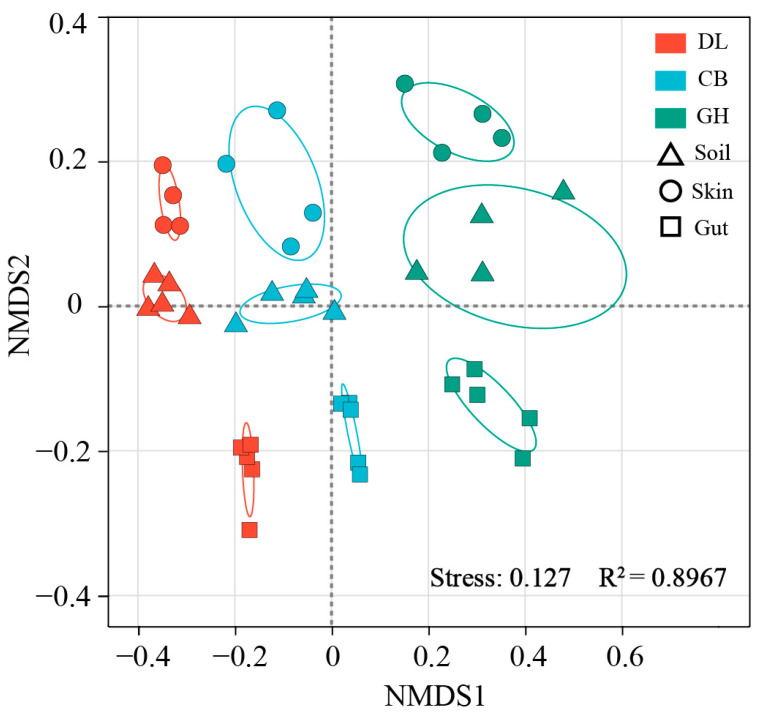
NMDS ranking plot based on Bray–Curtis dissimilarity in all samples. DL, Dongling Mountains Warm temperate deciduous broad-leaved forest; CB, Changbai Mountains Temperate broad-leaved red pine forest; GH, Greater Hinggan Mountains Cold temperate coniferous forest.

**Figure 7 microorganisms-12-01673-f007:**
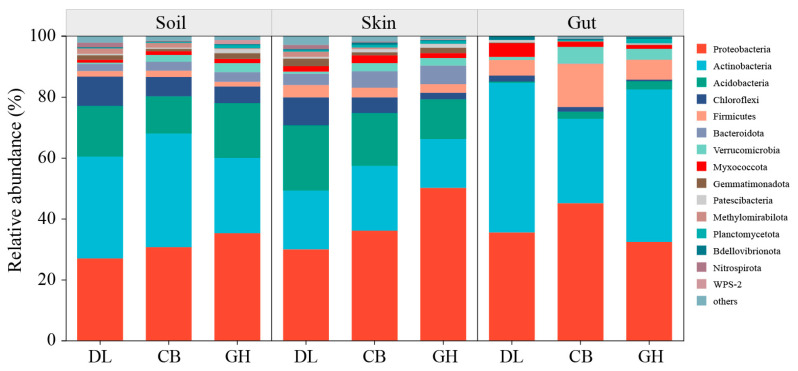
Bacterial community composition at the phylum level in three temperate forests. Taxa with relative abundances less than 1% were combined into “Others”. DL, Dongling Mountains Warm temperate deciduous broad-leaved forest; CB, Changbai Mountains Temperate broad-leaved red pine forest; GH, Greater Hinggan Mountains Cold temperate coniferous forest.

**Figure 8 microorganisms-12-01673-f008:**
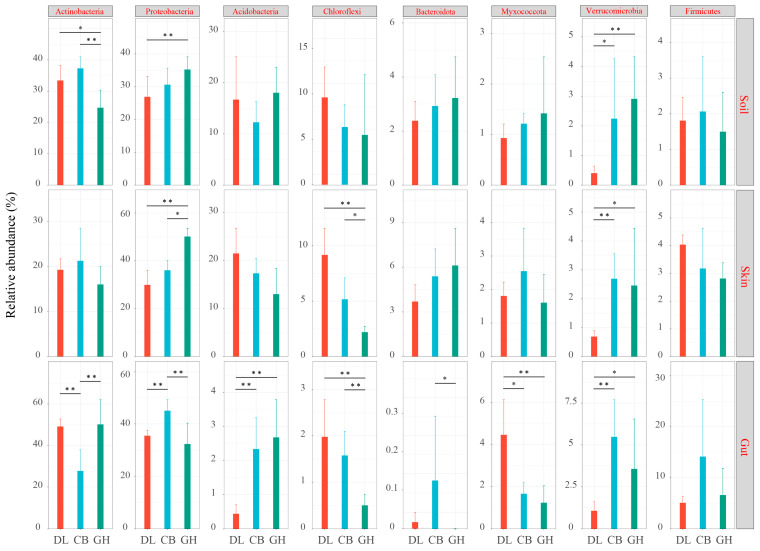
Variations in the relative abundance of dominant bacterial phyla across three temperate forests. DL, Dongling Mountains Warm temperate deciduous broad-leaved forest; CB, Changbai Mountains Temperate broad-leaved red pine forest; GH, Greater Hinggan Mountains Cold temperate coniferous forest. Significant differences are indicated by asterisks based on the nonparametric Kruskal–Wallis test: * *p* < 0.05 and ** *p* < 0.01.

**Figure 9 microorganisms-12-01673-f009:**
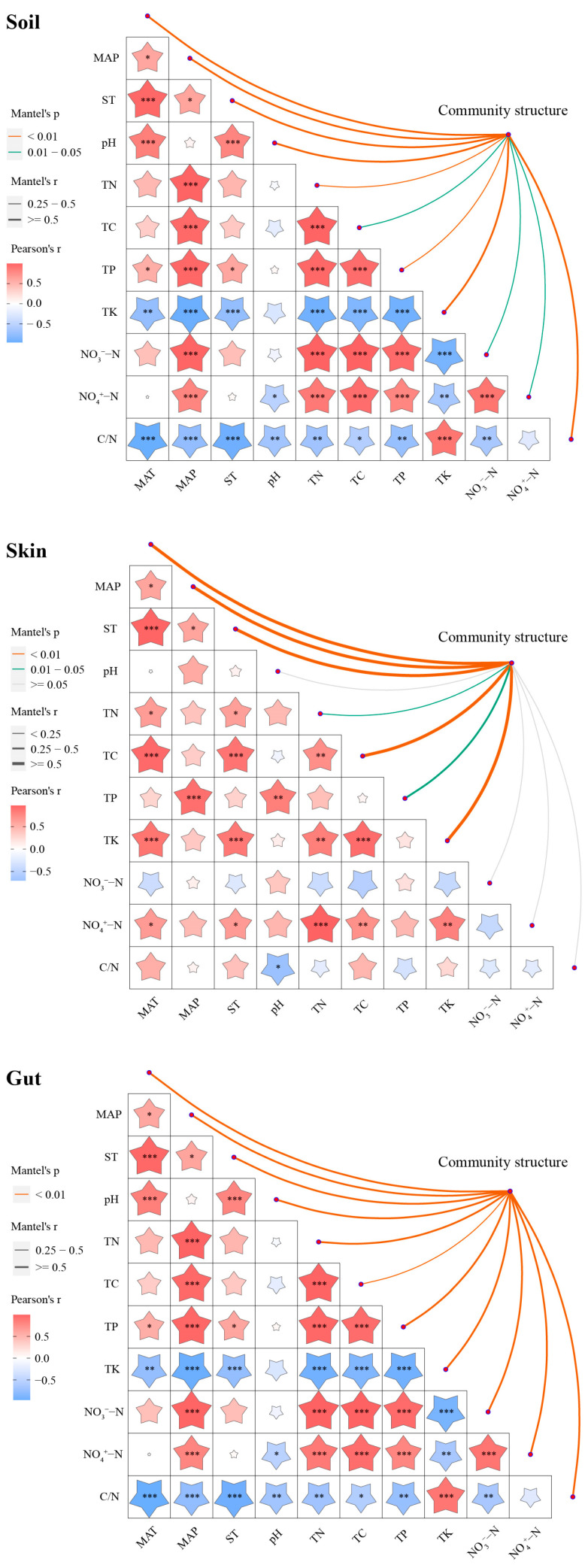
Mantel tests revealing the correlations between bacterial community structure and climatic factors, soil physicochemical properties, and physicochemical properties of earthworm epidermal mucus. MAT, mean annual temperature; MAP, mean annual precipitation; ST, soil temperature; TN, total nitrogen; TC, total carbon; TP, total phosphorus; TK, total potassium; NO_3_^−^-N, nitrate nitrogen; NH_4_^+^-N, ammonium nitrogen; C/N, carbon–nitrogen ratio. Different asterisks indicate significant differences at * *p* < 0.05, ** *p* < 0.01, and *** *p* < 0.001.

**Table 1 microorganisms-12-01673-t001:** ANOSIM and PERMANOVA analyses between the three temperate forests.

Group	Process	ANOSIM	PERMANOVA	
		R	R^2^	F
Soil	DL vs. CB	0.952 **	0.38007	4.9046 **
	DL vs. GH	0.996 **	0.40547	5.456 **
	CB vs. GH	0.832 **	0.32696	3.8864 **
Skin	DL vs. CB	0.776 *	0.32961	2.9501 **
	DL vs. GH	1 *	0.43064	5.1244 *
	CB vs. GH	0.896 *	0.3221	2.8508 *
Gut	DL vs. CB	1 **	0.59679	11.841 **
	DL vs. GH	1 **	0.50856	8.2787 **
	CB vs. GH	1 **	0.47142	7.1348 **

ANOSIM and PERMANOVA analyses based on Bray–Curtis dissimilarity were used to compare the bacterial community structure of earthworm habitat soil, skin, and gut in three temperate forests (permutation = 999). DL, Dongling Mountains Warm temperate deciduous broad-leaved forest; CB, Changbai Mountains Temperate broad-leaved red pine forest; GH, Greater Hinggan Mountains Cold temperate coniferous forest. Different asterisks indicate significant differences at * *p* < 0.05 and ** *p* < 0.01.

## Data Availability

All data are included in the article.

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
