# Peer review of "Comparative Analysis of Bacterial Community Structures in Earthworm Skin, Gut, and Habitat Soil across Typical Temperate Forests"

_microorganisms, 2024, doi:10.3390/microorganisms12081673_

Round 1

Reviewer 1 Report

Comments and Suggestions for Authors

The reviewed studies are important not only theoretically but can also be used practically in planning the development of forest areas. It would be worth considering the term environmental gradient. This statement is very general. I am not sure whether it can be used in the case of these studies. Usually, the environmental gradient is mentioned in the case of studies conducted in transects.

Lines 57-60: hypothesis ‘(1) Bacterial communities in earthworm skin, gut, and habitat soil will exhibit distinct response patterns to the environmental gradient from warm temperate deciduous broad-leaved forest to cold temperate coniferous forest’- the statement ‘environmental gradient’ is very general. Wouldn’t it be better to skip this statement and refer to forest types in the hypothesis, the proposal:’ ‘(1) Bacterial communities in earthworm skin, gut, and habitat soil will exhibit distinct response patterns from warm temperate deciduous broad-leaved forest to cold temperate coniferous forest’

Line 63: Study site – a map with marked research locations would be appreciated

Figures 1, 2, 3, 4, 5, 6, 7- please explain the abbreviations DL, CB, GH in the captions under figures; the same applies to Table 1. However, it would be worth considering whether the remaining abbreviations should also be explained in the captions and titles of figures and tables, this makes analyzing figures and tables much easier

Author Response

Comments 1:  The reviewed studies are important not only theoretically but can also be used practically in planning the development of forest areas. It would be worth considering the term environmental gradient. This statement is very general. I am not sure whether it can be used in the case of these studies. Usually, the environmental gradient is mentioned in the case of studies conducted in transects.

Lines 57-60: hypothesis ‘(1) Bacterial communities in earthworm skin, gut, and habitat soil will exhibit distinct response patterns to the environmental gradient from warm temperate deciduous broad-leaved forest to cold temperate coniferous forest’- the statement ‘environmental gradient’ is very general. Wouldn’t it be better to skip this statement and refer to forest types in the hypothesis, the proposal:’ ‘(1) Bacterial communities in earthworm skin, gut, and habitat soil will exhibit distinct response patterns from warm temperate deciduous broad-leaved forest to cold temperate coniferous forest’.

Response 1:  Thank you for raising these questions. We agree with this comment. Therefore, in the manuscript, we have removed 'environmental gradient' and made the following modifications:

(1) We made changes in lines 20-23 as follows:

Beta diversity analysis revealed that the transition from warm temperate deciduous broad-leaved forest to cold temperate coniferous forest exerted the most significant impact on the gut bacterial communities of earthworms, while its influence on the skin bacterial commu-nities was comparatively less pronounced.

(2) We made changes in lines 81-84 as follows:

Bacterial communities in earthworm skin, gut, and habitat soil will exhibit distinct re-sponse patterns from warm temperate deciduous broad-leaved forest to cold temperate coniferous forest;

(3) We made changes in lines 282-284 as follows:

The transition from DL to GH significantly influenced the bacterial community structures in earthworm skin, gut, and habitat soil (ANOSIM and PERMANOVA, p<0.05) (Table 1).

(4) We made changes in lines 284-288 as follows:

PERMANOVA analysis indicated that the transition from DL to GH exerted the most substantial influence on the bacterial community structure of earthworm gut (R2 = 0.471-0.596), followed by the earthworm habitat soil (R2 = 0.326-0.38), while the impact on the bacterial community structure of earthworm skin was comparatively less pronounced (R2 = 0.322-0.329) (Table 1).

(5) We made changes in lines 364-366 as follows:

We observed significant alterations in the diversity and structure of bacterial com-munities associated with earthworm skin, gut, and habitat soil along the transition from DL to GH.

(6) We made changes in lines 369-372 as follows:

Beta diversity analysis revealed disparities in bacterial community structures across different temperate forest ecosystems. Notably, the transition from DL to GH exhibited the most pronounced impact on the bacterial community structure within the earthworm gut.

Comments 2:  Line 63: Study site – a map with marked research locations would be appreciated.

Response 2:  Thank you for raising these questions. We agree with this comment. Therefore, we have added a map marked with the research locations in lines 104 to 107 of the manuscript. The details are as follows: 

Figure 1. Distribution map of the study plot. DL, Dongling Mountains Warm temperate decidu-ous broad-leaved forest; CB, Changbai Mountains Temperate broad-leaved red pine forest; GH, Greater Hinggan Mountains Cold temperate coniferous forest.

Comments 3:  Figures 1, 2, 3, 4, 5, 6, 7- please explain the abbreviations DL, CB, GH in the captions under figures; the same applies to Table 1. However, it would be worth considering whether the remaining abbreviations should also be explained in the captions and titles of figures and tables, this makes analyzing figures and tables much easier.

Response 3:  Thank you for raising these questions. We agree with this comment. Therefore, we have added the meanings of the abbreviations in Figures 1 to 9 and Table 1 of the manuscript.  The specific modifications are as follows:

(1) We have added the specific meanings of the abbreviations to Figure 1 in the manuscript, and this modification is located on lines 105 to 107 of the manuscript. The details of the changes are as follows:

Figure 1. Distribution map of the study plot. DL, Dongling Mountains Warm temperate deciduous broad-leaved forest; CB, Changbai Mountains Temperate broad-leaved red pine forest; GH, Greater Hinggan Mountains Cold temperate coniferous forest.

(2) We have added the specific meanings of the abbreviations to Figure 2 in the manuscript, and this modification is located on lines 225 to 230 of the manuscript. The details of the changes are as follows:

Figure 2. Soil physicochemical properties in three temperate forest types. DL, Dongling Mountains Warm temperate deciduous broad-leaved forest; CB, Changbai Mountains Temperate broad-leaved red pine forest; GH, Greater Hinggan Mountains Cold temperate coniferous forest; TN, total nitrogen; TC, total carbon; TP, total phosphorus; TK, total potassium; NO3--N, nitrate nitrogen; NH4+-N, ammonium nitrogen; C/N, carbon nitrogen ratio. Significant differences are indicated by asterisks based on the nonparametric Kruskal-Wallis test: *p<0.05, **p<0.01, and ***p<0.001.

(3) We have added the specific meanings of the abbreviations to Figure 3 in the manuscript, and this modification is located on lines 242 to 248 of the manuscript. The details of the changes are as follows:

Figure 3. Physicochemical properties of earthworm epidermal mucus in three temperate forest types. DL, Dongling Mountains Warm temperate deciduous broad-leaved forest; CB, Changbai Mountains Temperate broad-leaved red pine forest; GH, Greater Hinggan Mountains Cold temperate coniferous forest; TN, total nitrogen; TC, total carbon; TP, total phosphorus; TK, total potassium; NO3--N, nitrate nitrogen; NH4+-N, ammonium nitrogen; C/N, carbon nitrogen ratio. Significant differences are indicated by asterisks based on the nonparametric Kruskal-Wallis test: *p<0.05, **p<0.01, and ***p<0.001.

(4) We have added the specific meanings of the abbreviations to Figure 4 in the manuscript, and this modification is located on lines 254 to 258 of the manuscript. The details of the changes are as follows:

Figure 4. MAT, MAP, and ST in three temperate forest types. DL, Dongling Mountains Warm temperate deciduous broad-leaved forest; CB, Changbai Mountains Temperate broad-leaved red pine forest; GH, Greater Hinggan Mountains Cold temperate coniferous forest; MAT, mean annual temperature; MAP, mean annual precipitation; ST, soil temperature. Significant differences are indicated by asterisks based on the nonparametric Kruskal-Wallis test: *p<0.05, **p<0.01, and ***p<0.001.

(5) We have added the specific meanings of the abbreviations to Figure 5 in the manuscript, and this modification is located on lines 274 to 278 of the manuscript. The details of the changes are as follows:

Figure 5. Bacterial community α-diversity in earthworm habitat soil, skin, and gut across three temperate forest types. DL, Dongling Mountains Warm temperate deciduous broad-leaved forest; CB, Changbai Mountains Temperate broad-leaved red pine forest; GH, Greater Hinggan Mountains Cold temperate coniferous forest. Significant differences are indicated by asterisks based on the nonparametric Kruskal-Wallis test: *p<0.05, **p<0.01, and ***p<0.001.

(6) We have added the specific meanings of the abbreviations to Figure 6 in the manuscript, and this modification is located on lines 291 to 293 of the manuscript. The details of the changes are as follows:

Figure 6. NMDS ranking plot based on Bray-Curtis dissimilarity in all samples. DL, Dongling Mountains Warm temperate deciduous broad-leaved forest; CB, Changbai Mountains Temperate broad-leaved red pine forest; GH, Greater Hinggan Mountains Cold temperate coniferous forest.

(7) We have added the specific meanings of the abbreviations to Figure 7 in the manuscript, and this modification is located on lines 324 to 327 of the manuscript. The details of the changes are as follows:

Figure 7. Bacterial community composition at the phylum level in three temperate forests. Taxa with relative abundances less than 1% were combined into "Others". DL, Dongling Mountains Warm temperate deciduous broad-leaved forest; CB, Changbai Mountains Temperate broad-leaved red pine forest; GH, Greater Hinggan Mountains Cold temperate coniferous forest.

(8) We have added the specific meanings of the abbreviations to Figure 8 in the manuscript, and this modification is located on lines 329 to 333 of the manuscript. The details of the changes are as follows:

Figure 8. Variations in the relative abundance of dominant bacterial phyla across three temper-ate forests. DL, Dongling Mountains Warm temperate deciduous broad-leaved forest; CB, Changbai Mountains Temperate broad-leaved red pine forest; GH, Greater Hinggan Mountains Cold temperate coniferous forest. Significant differences are indicated by asterisks based on the nonparametric Kruskal-Wallis test: *p<0.05, **p<0.01, and ***p<0.001.

(9) We have added the specific meanings of the abbreviations to Figure 9 in the manuscript, and this modification is located on lines 349 to 353 of the manuscript. The details of the changes are as follows:

Figure 9. Mantel tests revealing the correlations between bacterial community structure and climatic factors, soil physicochemical properties, and physicochemical properties of earthworm epidermal mucus. MAT, mean annual temperature; MAP, mean annual precipitation; ST, soil temperature; TN, total nitrogen; TC, total carbon; TP, total phosphorus; TK, total potassium; NO3--N, nitrate nitrogen; NH4+-N, ammonium nitrogen; C/N, carbon nitrogen ratio.

(10) We have added the specific meanings of the abbreviations to Table 1 in the manuscript, and this modification is located on lines 297 to 300 of the manuscript. The details of the changes are as follows:

ANOSIM and PERMANOVA analyses based on Bray-Curtis dissimilarity were used to compare the bacterial community structure of earthworm habitat soil, skin, and gut in three temperate forests (permutation = 999). DL, Dongling Mountains Warm temperate deciduous broad-leaved forest; CB, Changbai Mountains Temperate broad-leaved red pine forest; GH, Greater Hinggan Mountains Cold temperate coniferous forest. Different asterisks indicate significant differences at *p<0.05 and **p<0.01.

Reviewer 2 Report

Comments and Suggestions for Authors The manuscript compares the composition and structure of bacterial communities in the skin, gut, and habitat soil of earthworm Eisenia nordenskioeldi across three typical temperate forests in China: warm temperate deciduous broad-leaved forest, temperate broad-leaved red pine forest, and cold temperate coniferous forest. The influence of climatic factors, and soil physicochemical properties was also investigated. The results are valuable and have practical meaning in terms of better understanding of the role and relations of earthworms in the environment. However, manuscript needs some improvements. Details are below  

Lines 255 – 256  (and then in the discussion section, lines 327 - 329) - " From DL to GH, the relative abundance of Verrucomicrobia significantly increased (p<0.05) in earthworm skin, gut, and habitat soil, while the relative abundance of Chloroflexi significantly decreased..” – this is too general – according to Figure 7 this tendency for Verrucomicrobia is seen only when we compare DL with GH (but not with CB) -  in the skin and gut CB is the highest). In the case of Chloroflexi  in soil differences are not statistically significant. Moreover, in the Figure 7 is Verrucomicrobiota while in the text is Verrucomicrobia – please, be consistent.

Lines 373 – 374 – „….and provides crucial evidence for further comprehending the impact of environmental changes on soil ecosystems.” – what evidence? This is overstatement – I agree with the first part of this sentence only  - I mean „This study deepens our understanding of earthworm-associated bacterial communities in temperate forest ecosystems”.

According to the references (which you have cited) in the discussion section there are many information about bacterial communities in soil, mucus and gut of earthworms. Please state clearly in the discussion section what is new (special) in your results.

Comments on the Quality of English Language

English fine

Author Response

Comments 1:  The manuscript compares the composition and structure of bacterial communities in the skin, gut, and habitat soil of earthworm Eisenia nordenskioeldi across three typical temperate forests in China: warm temperate deciduous broad-leaved forest, temperate broad-leaved red pine forest, and cold temperate coniferous forest. The influence of climatic factors, and soil physicochemical properties was also investigated. The results are valuable and have practical meaning in terms of better understanding of the role and relations of earthworms in the environment. However, manuscript needs some improvements. Details are below 

Lines 255 – 256  (and then in the discussion section, lines 327 - 329) - " From DL to GH, the relative abundance of Verrucomicrobia significantly increased (p<0.05) in earthworm skin, gut, and habitat soil, while the relative abundance of Chloroflexi significantly decreased..” – this is too general – according to Figure 7 this tendency for Verrucomicrobia is seen only when we compare DL with GH (but not with CB) -  in the skin and gut CB is the highest). In the case of Chloroflexi  in soil differences are not statistically significant.

Response 1:  Thank you for raising these questions. We agree with this comment. We have made revisions to the sections on 'Verrucomicrobia' and 'Chloroflexi', and the specific modifications are as follows:

(1) We made changes in lines 308-318 as follows:

In earthworm skin, gut, and habitat soil, both CB and GH demonstrated significantly higher relative abundances of Verrucomicrobia compared to DL (p<0.05). Specifically, CB exhibited a statistically significant increase in the relative abundance of Verrucomicrobia across all three microhabitats when compared to DL. GH displayed a similar trend relative to DL. However, no statistically significant differences (p>0.05) were observed between CB and GH in terms of Verrucomicrobia relative abundance in any of the examined microhabitats (earthworm skin, gut, or habitat soil). Compared to DL and CB, GH exhibited significantly lower relative abundances of Chloroflexi in both earthworm skin and gut (p<0.05). However, no statistically significant differences (p>0.05) were observed in the relative abundance of Chloroflexi in the earthworm habitat soil among the three temperate forest types (DL, CB, and GH).

(2) We made changes in lines 397-399 as follows:

For instance, compared to DL, both CB and GH exhibited significantly higher relative abundances of Verrucomicrobia in earthworm skin, gut, and habitat soil (p<0.05).

Comments 2:  Moreover, in the Figure 7 is Verrucomicrobiota while in the text is Verrucomicrobia – please, be consistent.

Response 2:  Thank you for raising these questions. We agree with this comment. We have made a correction in Figure 7, changing 'Verrucomicrobiota' to 'Verrucomicrobia'. 

Comments 3:  Lines 373 – 374 – „….and provides crucial evidence for further comprehending the impact of environmental changes on soil ecosystems.” – what evidence? This is overstatement – I agree with the first part of this sentence only  - I mean „This study deepens our understanding of earthworm-associated bacterial communities in temperate forest ecosystems”.

Response 3:  Thank you for raising these questions. We agree with this comment. We have removed 'and provides crucial evidence for further comprehending the impact of environmental changes on soil ecosystems' and revised the sentence to 'This study deepens our understanding of earthworm-associated bacterial communities in temperate forest ecosystems'.

We made changes in lines 442-443 as follows:

This study deepens our understanding of earthworm-associated bacterial communities in temperate forest ecosystems.

Comments 4:  According to the references (which you have cited) in the discussion section there are many information about bacterial communities in soil, mucus and gut of earthworms. Please state clearly in the discussion section what is new (special) in your results.

Response 4:  Thank you for your valuable suggestions. We have carefully considered your advice regarding the need to clearly articulate the novelties of our study in the discussion section. Below, we outline the specific novelties of our research:

(1) Uniqueness of Research Scope: Our study conducted a comparative analysis of the bacterial communities in three microhabitats of earthworms – skin, gut, and habitat soil – across three typical temperate forests in China. This comprehensive comparison across different geographical regions and multiple microhabitats offers a new perspective on understanding the ecological role of earthworms.

(2) Differential Responses to Environmental Gradients: Our findings reveal that the transition from warm temperate deciduous broad-leaved forests to cold temperate coniferous forests has significantly different impacts on bacterial communities in various microhabitats, with the gut bacterial community being most affected. This discovery of a differentiated response pattern to environmental changes represents an important innovation in our study.

To better reflect these novelties, we have revised the " 4. Discussion " section accordingly and have explicitly pointed out these contributions at the following line numbers: 362-363, 373-374, 377-379, and 403-410.

We hope that these revisions meet your suggestions and further enhance the contributions of our research. Thank you once again for your insightful review.

The specific revisions are as follows:

  1. Discussion

4.1. Effects of Temperate Forest Type Transition on Bacterial Community Diversity in Earthworm-Associated Microhabitats

We observed significant differences in the physicochemical properties of earthworm habitat soil and skin mucus across the three temperate forest types. These variations are likely driven by changes in climatic factors and vegetation types [36]. The physicochemical properties of earthworm skin mucus exhibited different trends compared to those of habitat soil, suggesting that earthworms may adapt to diverse environmental conditions by regulating the secretion of skin mucus [37, 38]. This discovery provides novel insights into the adaptive mechanisms of earthworms in diverse temperate forest ecosystems.

We observed significant alterations in the diversity and structure of bacterial communities associated with earthworm skin, gut, and habitat soil along the transition from DL to GH. Alpha diversity analysis revealed a gradual decrease in bacterial community diversity from DL to GH, potentially due to reduced temperature and nutrient availability limiting microbial growth and metabolism [39].

Beta diversity analysis revealed disparities in bacterial community structures across different temperate forest ecosystems. Notably, the transition from DL to GH exhibited the most pronounced impact on the bacterial community structure within the earthworm gut. This indicates that earthworm gut bacterial communities are more sensitive to environmental changes compared to those in habitat soil and skin. This differentiated response pattern represents a novel finding. This heightened sensitivity may be attributed to the close association between gut bacterial communities and the host's digestive and nutrient absorption processes [40], their higher metabolic capacity [12, 41], and potential influences from changes in food sources [42]. This discovery expands our understanding of earthworm-microbe-environment interactions, underscoring the potential value of earthworm gut microbiomes as indicators of environmental change.

4.2. Impact of Temperate Forest Type Transition on Bacterial Community Composition in Earthworm-Associated Microhabitats

Regarding bacterial community composition, we found that Actinobacteria and Proteobacteria were the predominant phyla in earthworm skin, gut, and habitat soil [43, 44]. The bacterial community composition in earthworm gut microbiomes exhibited significant differences compared to those in earthworm skin and habitat soil. Acidobacteria and Chloroflexi demonstrated higher relative abundances in earthworm skin and habitat soil compared to the earthworm gut [45]. Acidobacteria, Actinobacteria, and Proteobacteria were found to be predominant in forest soils [46-48]. The higher carbon-to-nitrogen ratio in earthworm guts resulted in substantially lower abundances of Acidobacteria compared to habitat soil [49]. Firmicutes and Verrucomicrobia displayed higher relative abundances in earthworm guts compared to earthworm skin and habitat soil. These phyla are recognized as dominant taxa in earthworm gut microbiomes [12, 41, 49-53]. The earthworm gut provides a near-anaerobic microenvironment [33, 54], which favors the growth of anaerobic or facultative anaerobic bacteria [55]. Certain bacteria within the Firmicutes are well-adapted to thrive in such conditions [56].

Significant changes in the relative abundances of certain bacterial phyla were observed across the gradient from DL to GH. For instance, compared to DL, both CB and GH exhibited significantly higher relative abundances of Verrucomicrobia in earthworm skin, gut, and habitat soil (p<0.05). This trend may be attributed to the preference of Verrucomicrobia for oligotrophic environments [57]. Previous studies have shown that the abundance of Actinobacteria in acidic soils positively correlates with pH [58, 59]. The gradual decrease in pH of earthworm habitat soil from DL to GH may explain the reduction in the relative abundance of Actinobacteria in this microhabitat. From DL to GH, the relative abundance of Acidobacteria in earthworm gut showed a gradually increasing trend, while there was no significant difference in its relative abundance in earthworm skin and habitat soil. Acidobacteria are considered slow-growing oligotrophs [60] capable of utilizing complex carbon substrates [61]. The increase in the relative abundance of Acidobacteria in the earthworm gut may reflect changes in the host's digestive strategy, allowing Acidobacteria to gain an advantage in the gut of GH earthworms and thus play a more important role in the host's digestive process.

Reviewer 3 Report

Comments and Suggestions for Authors

Comparative Analysis of Bacterial Community Structures in Earthworm Epidermis, Gut, and Habitat Soil Across Typica Temperate Forests

Abstract

Use as keywords significant words, but not the ones in title.

Introduction

It is too short. It needs more specific details on how earthworm-soil-environment are dependent on mutually, what are the action of earthworm on soil, etc. So, the reader can understand the relevance of the study. To say that this remains unclear is not enough.

Methodology

It is necessary to indicate that all different studies and determinations, on earthworms and soils, were carried out for the three different forest ecosystems.

Results

Present results in the same order that are described in methodology.

Conclusions

What is presented in collusion is a summary of results.

To write conclusions focus on the hypothesis presented at the end of introduction.

General comment

Was in the soil only one type of earthworm?

Why to study only one? This needs support.

Author Response

Comments 1:  Abstract

Use as keywords significant words, but not the ones in title.

Response 1:  Thank you for raising these questions. We agree with this comment. In response to your feedback, we have revised the keywords section. The updated keywords are now: " earthworm; bacteria; diversity analysis; mantel test; environmental factors " This change has been made at line 30 of the manuscript.

Keywords: earthworm; bacteria; diversity analysis; mantel test; environmental factors

Comments 2:   Introduction

It is too short. It needs more specific details on how earthworm-soil-environment are dependent on mutually, what are the action of earthworm on soil, etc. So, the reader can understand the relevance of the study. To say that this remains unclear is not enough.

Response 2:  Thank you for your insightful comments regarding the Introduction section of our manuscript. We agree with your assessment that it was too brief and needed more specific details to clarify the interdependencies among earthworms, soil, and the environment, as well as the actions of earthworms on soil.

In response to your feedback, we have thoroughly revised the "1 Introduction" section to include a more detailed explanation of how earthworms, soil, and the environment are interdependent and interact with each other. We have also elaborated on the specific roles that earthworms play in soil processes. These revisions span from line 33 to line 75 of the manuscript.

  1. Introduction

Earthworms, functioning as key ecosystem engineers in soil ecosystems, play a crucial role in maintaining soil health and promoting ecosystem functions [1]. These invertebrates exhibit complex interdependencies with soil and environmental factors, which are essential for understanding and managing forest ecosystems [2].

The impact of earthworms on soil is multifaceted and can be categorized into direct and indirect effects. Direct influences are primarily achieved through their feeding and excretion activities. Earthworms ingest soil and organic matter, mixing and enriching nutrients during digestion, and subsequently excreting them in the form of earthworm casts [3]. This process not only accelerates the decomposition of organic matter but also enhances the bioavailability of soil nutrients [4]. For instance, earthworm activity can significantly increase the content of key elements such as nitrogen, phosphorus, and potassium in the soil [5]. Furthermore, the burrowing and tunneling behaviors of earthworms improve the physical structure of the soil [6, 7]. The network of channels they create in the soil increases soil porosity, facilitating the circulation of water and air, thereby enhancing soil aeration and permeability [8]. These modifications to soil structure are conducive to plant root growth [9]. The indirect effects of earthworms are primarily manifested through their complex interactions with microbial communities. The diverse and abundant microbial populations present in earthworm skin and gut play crucial roles in processes such as organic matter decomposition, nutrient transformation, and pathogen suppression [10-13]. For instance, microorganisms in the earthworm gut secrete various enzymes that accelerate the degradation of recalcitrant organic compounds such as cellulose and hemicellulose [14]. Concurrently, earthworm activity can alter the composition and function of soil microbial communities, thereby influencing the biogeochemical cycles of the entire soil ecosystem [15, 16]. However, the activity and distribution of earthworms are also strongly influenced by environmental factors. Climatic conditions (such as temperature and precipitation), soil physicochemical properties (including pH and organic matter content), and vegetation types directly or indirectly affect earthworm survival, reproduction, and activity [17-21]. This interplay creates a complex feedback loop, rendering the earthworm-soil-environment system an intricately interconnected entity.

Temperate forests, as one of the most widely distributed terrestrial ecosystems on Earth, play an irreplaceable role in global carbon cycling, biodiversity conservation, and climate regulation [22, 23]. However, different types of temperate forests exhibit significant variations in climatic conditions, vegetation composition, and soil characteristics, which may profoundly influence earthworms and their associated microbial communities [24]. Concurrently, the quality and quantity of litter in various forest types affect soil organic matter content and quality [25, 26], subsequently impacting earthworm food sources and microbial community composition. Although previous studies have explored the distribution patterns and ecological functions of earthworms in temperate forests [27-29], systematic comparative research on soil, skin, and gut microbial communities associated with earthworms across different temperate forest types remains limited. In particular, our understanding of how the transition from warm temperate deciduous broadleaf forests to cold temperate coniferous forests influences earthworm-associated microbial communities is still constrained.

Comments 3:  Methodology

It is necessary to indicate that all different studies and determinations, on earthworms and soils, were carried out for the three different forest ecosystems.

Response 3:  Thank you for raising these questions. We agree with this comment. In response to your feedback, we have made the necessary modifications to the Methodology section. Specifically, we have added a statement at lines 101 to 103 to indicate that "All studies and determinations pertaining to earthworm skin, gut, and habitat soil were conducted across the three distinct forest ecosystems."

Comments 4:  Results

Present results in the same order that are described in methodology.

Response 4:  Thank you for raising these questions. We agree with this comment. We have adjusted the order within the "2 Materials and Methods" section to ensure a direct correspondence with the "3 Results" section. This involved revisions at lines 86-87, 92-93, 108, 123, 137, 151, 158, 171, and 193 of the manuscript. The revised order in "2 Materials and Methods" is as follows:

2. Materials and Methods

   2.1.  Ethics Statement

   2.2.  Sample Collection

      2.2.1.  Study Site

      2.2.2.  Experiment Design

      2.2.3.  Bacterial Collection from Earthworm Skin

      2.2.4.  Earthworm Skin Mucus Collection

      2.2.5.  Earthworm Gut Collection

  2.3.  Physicochemical Properties of Soil and Skin Mucus

  2.4.  DNA Extraction of Bacterial Communities

  2.5.  Statistical Analyses

Comments 5:    Conclusions

What is presented in collusion is a summary of results.

To write conclusions focus on the hypothesis presented at the end of introduction.

Response 5:  Thank you for your insightful comment regarding the Conclusions section of our manuscript. We agree with this comment. We have rewritten the Conclusions section to directly address the two main hypotheses introduced in the Introduction. The revised conclusions, which can be found at lines 433 to 443, now explicitly state the following:

This study compared the earthworm-associated bacterial communities in three typical temperate forests in China and validated our two main hypotheses: (1) The transition from warm temperate deciduous broad-leaved forest to cold temperate coniferous forest significantly influenced the bacterial community structure in earthworm skin, gut, and habitat soil, but the extent of the impact varied among microhabitats. The earthworm gut bacterial community was the most affected, while the skin bacterial community was relatively less affected. (2) These differences were primarily influenced by climatic factors, soil physicochemical properties, and the properties of earthworm skin mucus. Mantel test results indicated that these three types of factors were significantly correlated with the bacterial community structure in different microhabitats. This study deepens our under-standing of earthworm-associated bacterial communities in temperate forest ecosystems.

Comments 6:  General comment

Was in the soil only one type of earthworm?

Why to study only one? This needs support.

Response 6:  Thank you for your general comment regarding the choice of earthworm species in our study. We appreciate your attention to the ecological diversity and the importance of studying multiple species. In response to your query, we would like to provide additional context and justification for our focus on Eisenia nordenskioldi. The specific explanation is as follows:

Eisenia nordenskioldi is a widely distributed cold-tolerant earthworm species, predominantly inhabiting the soil and leaf litter layers in Siberia and the northeastern regions of China. This species is found across a broad range of ecosystems in Asia and Eastern Europe, including tundra, boreal forests, and forest-steppe zones. Eisenia nordenskioldi is a dominant species in many tundra and cold habitats, playing a crucial role in maintaining soil health and nutrient cycling. As such, it is a representative earthworm species in the tundra and cold environments of Northeast Asia (References [1-3]).

We found Eisenia nordenskioldi in all three temperate forest study sites. To further confirm the distribution of earthworm species in this region, we conducted extensive surveys within the 25 hm2 forest monitoring plot in the Huzhong National Nature Reserve of Heilongjiang Province and the surrounding areas of Huzhong Town, Huzhong District, Daxing'anling Region of Heilongjiang Province. Our results revealed the presence of only this single earthworm species. We also consulted with local residents and Professor Yufeng Zhang from the Hebei Key Laboratory of Animal Diversity at Langfang Normal University, who confirmed that Eisenia nordenskioldi might be the only earthworm species distributed in this region. Furthermore, upon referring to the book "Terrestrial Earthworms of China" co-authored by Qin Xu and Nengwen Xiao, we found that only this single earthworm species was recorded in some parts of Heilongjiang Province (Reference [4]).

Although other earthworm species were found in the other two study sites, we ultimately decided to select Eisenia nordenskioldi as our research object to ensure consistency and comparability across our study. By comparing the differences of the same species under various habitat conditions, we can better reveal the influence of environmental factors on the earthworm-associated bacterial communities without the interference of species differences.

References:

[1] Xiao, T., Zhang, B., Zhao, H., Xie, Z., Zhang, Y., Wu, D., Chen, T.-W., Scheu, S., Schaefer, I., 2024. Differential changes in body size and stoichiometry in genetic lineages of the earthworm Eisenia nordenskioldi with elevation. Soil Biology and Biochemistry 189, 109262. https://www.sciencedirect.com/science/article/pii/S0038071723003243

[2] Hao, J., Wang, L., Aspe, N.M., Han, A.C., Chen, M., Li, M., Zhang, S., Wu, D., 2024. Seasonal dynamics of gut microbiome: A study of multi-kingdom microbiota of earthworm gut in an urban park. Applied Soil Ecology 195, 105259. https://doi.org/10.1016/j.apsoil.2023.105259

[3] Zhang, Y., Zhang, Y., Wu, H., Li, C., Aspe, N.M., Wu, D., 2022. Patterns of Genetic Variation in the Eisenia nordenskioldi Complex (Oligochaeta: Lumbricidae) along an Elevation Gradient in Northern China. Diversity 14, 35. https://doi.org/10.3390/d14010035

[4] Xu Qin, Xiao Nengwen. Terrestrial Earthworms in China [M]. Beijing: China Agriculture Press, 2010.
